# Impact of occupational stress on healthcare workers' family members before and during COVID-19: A systematic review

**Sahra Tekin** [1]*, **Helen Nicholls**[1], **Dannielle Lamb**[2], **Naomi Glover**[1], **Jo Billings** [1]

**1** Division of Psychiatry, University College London, London, United Kingdom, **2** Department of Applied Health Research, University College London, London, United Kingdom

* sahra.tekin.20@ucl.ac.uk

## Abstract

We aimed to explore the experiences, needs, and mental health impact of family members of healthcare workers (HCWs) before and during the COVID-19 pandemic. Eleven quantitative and nine qualitative studies were included in this review. Based on the narrative synthesis, we identified five outcomes: 'Mental health outcomes', 'Family relationships, 'Coping skills and resilience', 'Quality of life and social life', and 'Practical outcomes'. Our findings indicated that there was a high risk to the mental health and well-being of families of healthcare workers both before and during the pandemic. However, with the COVID-19 pandemic, some experiences and mental health issues of families were worsened. There was also a negative association between working long hours/shift work and family relationships/communication, family social life, and joint activities, and family members taking on more domestic responsibilities. Families tended to use both positive and negative coping strategies to deal with their loved one's job stress. Organisations and support services working with people in health care work should consider widening support to families where possible. With this understanding, HCWs and their families could be supported more effectively in clinical and organisational settings.

**Trial registration: Systematic Review Registration Number**: CRD42022310729. https://www.crd.york.ac.uk/prospero/display_record.php?ID=CRD42022310729.

**Data Availability Statement:** The data contained in this review is already available in the public domain

## 1. Introduction

There were almost 1.3 million healthcare workers (HCWs) in the UK in 2023 [1], and in 2020, 65.1 million HCWs worldwide [2]. Due to the nature of healthcare work and associated risk of exposure to traumatic stress such as high rates of morbidity and mortality of patients [3], HCWs are at risk of developing mental health issues such as depression, anxiety, and PTSD [4]. Occupational stress amongst HCWs long preceded the COVID-19 pandemic, for example, a pre-COVID-19 review study, found HCWs experienced burnout, distress, anxiety, and psychosomatic symptoms [5]. Similarly, an earlier study found that HCWs who experienced burnout reported lower self-rated physical health (such as back and neck pain), greater sleep disturbances and impaired memory [6].

and the authors provided DOI numbers of included studies in the References.

**Funding:** The author(s) received no specific funding for this work.

**Competing interests:** NO authors have competing interests.

Similar findings have been seen since the COVID-19 pandemic, with a systematic review of experiences of HCWs during the COVID-19 and previous pandemics showing that long working hours, limited resources and unsocial shifts were significantly challenging for HCWs' psychosocial wellbeing [4]. Long and inflexible working hours, unsafe or poor working conditions, low pay, and limited support from colleagues and supervisors have been shown to increase the risk of mental health issues at work [7]. Based on recent literature, the prevalence of occupational PTSD among emergency medical service (EMS) workers who have experienced work-related trauma, is estimated to be 8.4–41.1%, although estimates vary due to differences in the description of PTSD, type of traumatic event, exposure period, and differences in occupation [8].

Research has consistently shown that social support is one of the key protective factors against the development of PTSD [9,10]. Workers from high-risk jobs may seek support from their families. However, this support can come at a cost. Exposure to trauma at work and PTSD impact not only the mental health and well-being of individuals exposed when those individuals come back home and share their traumatic work experiences with their families, this may also affect their families negatively [11].

As yet, there has been relatively little research into the impact of occupational stress on HCWs families and no previous synthesis of what literature is available. Wider literature, however, highlights the difficulties that can be experienced by families of other high-risk workers such as police officers and firefighters. Findings of a recent systematic review of the experiences of families of emergency responders with PTSD highlight that families of emergency responders with PTSD can experience vicarious and secondary trauma [12]. Researchers reported that spouses of first responders were overwhelmed because of the increased domestic responsibilities and their new "carer responsibilities" [13]. Similarly, spouses of law enforcement officers reported that they may experience nausea, intrusive thoughts, anxiety, and physiological symptoms such as shaking, after listening to what their law enforcement spouses had experienced after a traumatic event at work [14]. Regehr et al., [15] highlighted that spouses of firefighters were keen to support their firefighter partners psychologically, but that this had a negative impact on their own wellbeing and increased their worry. According to Uchida et al., [16], children of World Trade Centre responders in 2001, tended to experience behavioural problems such as fearful/clingy behaviours.

There is, to date, little research on HCWs at high-risk of being exposed to trauma at work, and very little consideration of their families, despite the consistently demonstrated benefit of familial social support, and potentially detrimental impact of occupational stress on families. In this review, we aimed to explore the impact of occupational stress and exposure to trauma on HCWs' families by systematically reviewing existing primary research and synthesising findings across the literature. Additionally, we aimed to provide insight into the experiences and mental health of families of HCWs before the COVID-19 pandemic and during/after the pandemic.

## 2. Method

### 2.1. Study design and registration

The systematic review protocol was registered on the NIHR's International Prospective Register of Systematic Reviews (PROSPERO) with the registration number "CRD42022310729". We adhered to PRISMA (Preferred Reporting Items for Systematic Reviews and Meta-analyses) guidance throughout this review [17].

**Table 1. Key search terms.**

| Sample | Phenomenon of Interest | Design | Evaluation | Research type |
|---|---|---|---|---|
| • High-risk occupational groups<br>• Family members<br>• Family relationships | • Occupational trauma<br>• Occupational stress | • Survey<br>• Interview | • Vicarious trauma<br>• Experiences<br>• Views<br>• Family satisfaction<br>• Interpersonal relationships | Original empirical peer reviewed published research, including quantitative, qualitative and mixed methods studies. |

## 2.2. Search strategy

We conducted a systematic literature search using the following electronic databases: Medline (Ovid), PTSDpubs, PsychINFO(Ovid), EMBASE(Ovid), and Scopus. Initial literature searches were completed between July 2022 and August 2022 and updated between August and September 2023.

Key words related to the research questions were organised based on the SPIDER tool. Alternative terms were detailed to include database-specific topic titles and Medical Subject Headings. The key search terms are listed in Table 1. (For the full list of search terms see Supporting Information 1 in S1 File. The results from the database searches were imported to reference management software EndNoteX9, and duplicates were removed. Backwards and forwards citation searching of included papers was also conducted to identify other potentially relevant papers.

## 2.3. Eligibility criteria

Articles were included based on following criteria: a) peer-reviewed published qualitative, quantitative, or mixed method studies written in English or Turkish; b) either comprised of a sample which identified its population as HCWs who talk about their families' experiences, needs, mental health, wellbeing, and/or their family life, or comprised of a sample which identified its population as families of HCWs; c) research that focused sufficiently on the impact of occupational stress on families of HCWs in terms of family life (family relationship, family cohesion, interpersonal relationships, family and social support), mental health (vicarious trauma, secondary trauma, post-traumatic stress disorder, stress-related disorders, compassion fatigue, burnout) and/or wellbeing of families (coping, happiness, marriage satisfaction, domestic responsibilities, impact of work schedule and shifts), and their needs and experiences as families of those in healthcare work.

Articles were excluded if: a) they did not focus on the HCWs' family members' mental health, wellbeing and/or experiences; b) they did not focus on the impact of occupational stress experienced by HCWs on their families; c) studies were related to other high-risk occupational group workers' families; d) they were written before 1980.

We excluded studies prior to 1980 due to PTSD first being recognised as a diagnosis in the DSM III in 1980, and to capture more relevant research on the nature of modern working across the last 40 years.

## 2.4. Data extraction and quality appraisal

The following information was extracted where available: Authors, date of publication, country, study design, type of qualitative/quantitative analyses used, sample size, (if specified)

HCWs' role, relationship with HCW, and main findings, including themes identified in the qualitative and mixed methods research.

We appraised the quality of studies using the Critical Appraisal Skills Programme (CASP) checklist [18] for qualitative studies and Appraisal tool for Cross-Sectional Studies (AXIS) [19] for cross-sectional studies. Additionally, AMSTAR checklist [20] was used to evaluate the quality of this review and the results showed that this review is a high-quality review (See Supporting Information 2 in S1 File).

## 2.5. Synthesis

In this review we have used narrative synthesis to organise our findings. Neither meta-analysis nor meta-synthesis were applicable for this study because of the wide variability of studies in relation to study design, types of relationships between family members and HCWs, and outcome measures. The evidence was narratively synthesised by following Popay et al.'s [21] approach. According to Popay et al., [21], there are four main elements in a narrative synthesis:

a)***Developing a theoretical model***: In our review study, we determined our research questions, and we provided information regarding the inclusion and exclusion criteria to address this element.

b) ***Developing a preliminary synthesis***: In this stage, the aim is to provide preliminary findings of the included studies. Popay et al., [21] point out different tools and techniques during stage. In this review, we preferred to use "translating data; thematic analysis" [22,23], because we aimed to examine the findings of both qualitative and quantitative studies focusing on the experiences and mental health of healthcare professionals' families in terms of the similarities and differences. A list of potential preliminary codes and themes were generated from the findings by ST. At research meetings, these preliminary codes and themes were discussed based on the feedback from the research team, themes were improved, and final themes were determined.

c) ***Exploring relationships in the data***: In order to explore the relationship in the findings, we used a conceptual mapping technique [24]. In this stage, ST re-read all the themes and findings of the included studies and compared and contrasted them based on their similarities and differences.

d) ***Assessing the robustness of the synthesis***: According to Popay et al., [10] for robustness, the quality of the included studies and the trustworthiness of the synthesis are significant. In order to assess the quality of the included studies and enhance the trustworthiness of the review; we completed quality appraisals for each included study. To minimise bias, all researchers were included in different stages. Two researchers (ST and HN) independently completed the title/abstract and full-text screening. During the synthesis, ST analysed the data and discussed the results with JB, and NG and DL re-read the manuscript and provided feedback.

## 3. Results

### 3.1. Study selection

From database searches, we identified 16,984 articles (from July-August 2022 search) and 2345 articles (from September 2023 search). After deduplication, the abstracts and titles of 14,332 articles were screened by ST, and a subset (N = 700) were independently screened by HN. We excluded 14,099 articles that were not relevant to the research questions. Based on our eligibility criteria, ST completed full-text screening of 233 articles and HN independently reviewed 40 articles. At this stage, 218 articles were excluded for the following reasons "not related to

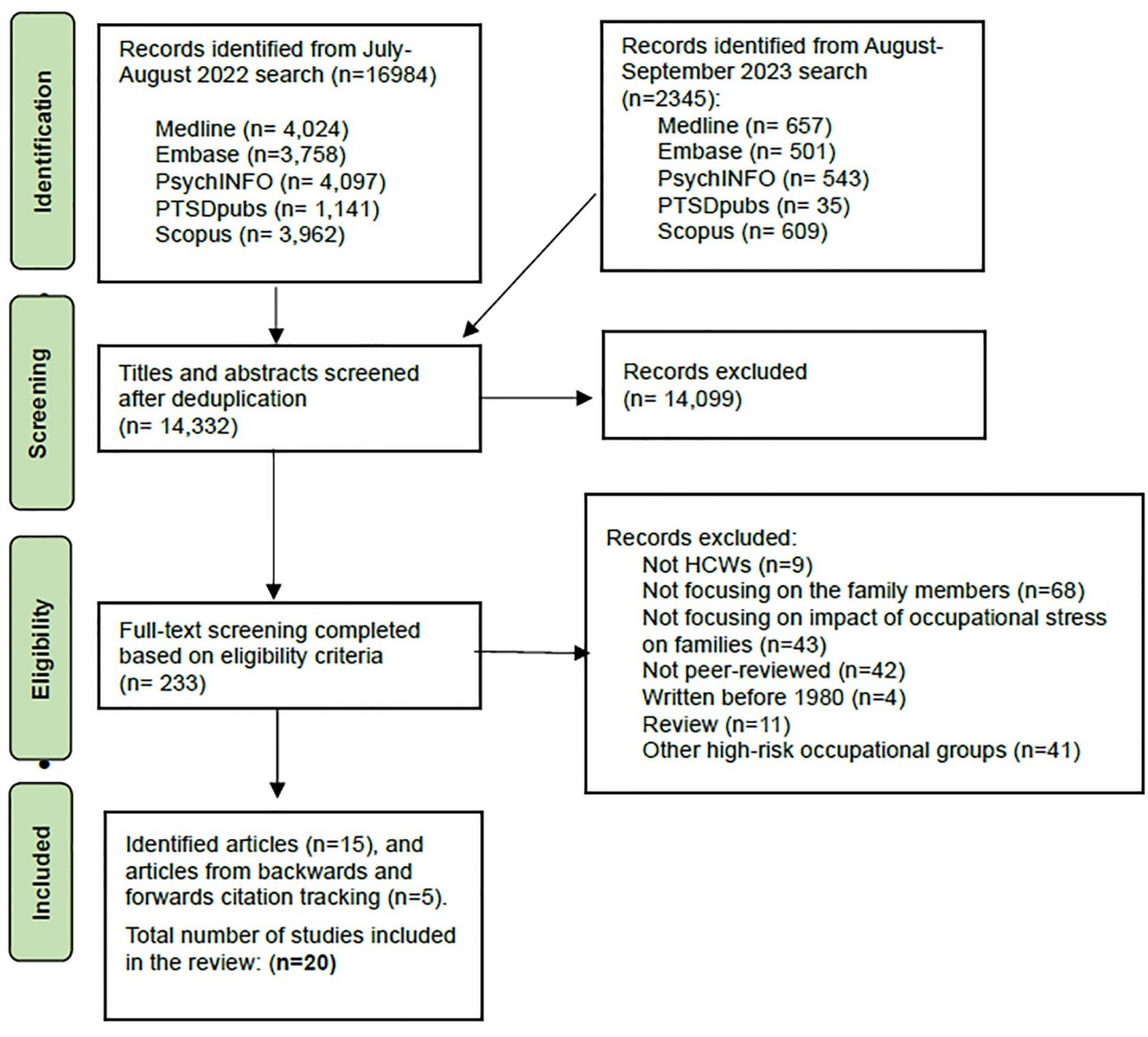

**Fig 1. PRISMA Flow chart of study selection.**

HCWs ($n = 9$)", "not focusing on families ($n = 68$)", "not focusing on the impact of occupational stress on family members ($n = 43$)", "not peer-reviewed ($n = 42$)", "written before 1980 ($n = 4$)", "review studies ($n = 11$)", and "related to other high-risk occupational group workers and/or their families ($n = 41$). An additional 5 records were identified through backwards and forwards citation tracking. In total, 20 articles were included in our review. Fig 1. shows a PRISMA Flowchart of the process of the screening and selecting included studies. (See Fig 1. PRISMA Flow chart of study selection, and see Supporting Information 3 for PRISMA Checklist in S1 File).

### 3.2. Study characteristics

Study and sample characteristics of the included quantitative and qualitative studies are shown in Tables 2 and 3, respectively. Eleven studies were quantitative designs [25–35] and

**Table 2. Characteristics of the included quantitative studies.**

| First Author (Year) | Location | Study Design | Healthcare Worker Group | Study Population | Research Aims | Main Findings |
|---|---|---|---|---|---|---|
| Banitalebi, et al., (2021) [28] | Iran | Cross-Sectional | Nurses | Family members (n = 208) | Assessing the impact of COVID-19 pandemic on mental health of family members of nurses | • High prevalence of depression symptoms experienced by family members<br>• Potential risk factors for family members of nurses included age, gender, marriage status |
| Banitalebi, et al., (2022) [35] | Iran | Cross-sectional | Nurses | Family members (n = 220) | Investigation of the association between coping skills with mental health and quality of life of the family members of nurses during the COVID-19 pandemic | • Higher coping skills score was related to higher scores in mental health and quality of life |
| Chua, et al., (2021) [33] | Hong Kong | Cross-Sectional | Healthcare Workers (HCW) (Doctors, Nurses, Dentists, Pharmacists, Allied health professionals, and Healthcare assistants) | HCWs (n = 747) and families (n = 245 | Assessing the HCWs perceived stress and its impact of family members and family relationships | • Higher perceived stress of HCWs was related with negative changes in family relationships. However, there was a positive association between perceived stress of HCWs and family cohesion and family members stress level. |
| Feng, et al., (2020) [29] | Chine | Cross-Sectional | Frontline rescue workers (93 front-line doctors, 179 nurses, 31 medical technicians, 82 rear-service personnel, 99 community street inspection personnel, 20 cleaning staff, 78 volunteers, 30 police, and 59 managers | Family members (n = 671) | Exploring the psychological distress of family members of frontline rescue workers | • Mental health outcomes of the frontline work for family members such as sleep problems, anxiety, depression, PTSD.<br>• Increased domestic responsibilities at the home such as childcare, elderly care, dealing with the daily life issues |
| Goud et al., (2021) [26] | India | Cross-Sectional | Doctors | Spouses (n = 120) | Investigating the level of psychological distress which was experienced by spouses of doctors, and factors which contributes to enhancement of this stress level. | • High psychological distress was reported by spouses<br>• Financial insecurity was a risk factor for psychological distress |
| Henry et al., (2023) [32] | US | Cross-sectional | Emergency Medical Service personnel (EMS) (Emergency medical technician or paramedics who work for an ambulance service) | Emergency Medical Service personnel and their partners (n = 30 couples) | Assessing the association between EMS workers' self-reported PTSD symptoms and EMS workers' partners' self-reported PTSD symptoms, relationship satisfaction, and social support | • There was an increased risk for partners of EMS workers who met diagnostic criteria for PTSD to develop PTSD symptoms compared to partners of EMS workers who met one criteria or no criteria of PTSD.<br>• There was a negative impact of EMS workers' PTSD symptoms on partners' satisfaction for social support. |

*(Continued)*

**Table 2.** (Continued)

| First Author (Year) | Location | Study Design | Healthcare Worker Group | Study Population | Research Aims | Main Findings |
|---|---|---|---|---|---|---|
| King & DeLongis (2014) [34] | Canada | Longitudinal | Paramedics | Paramedics and their spouses (*n* = 87 couples) | Investigation of the occupational stress related negative coping skills such as rumination and interpersonal withdrawal and their impact on the relationships with their spouses. | • There was a relationship between paramedic's perceived stress and burnout experiences at work and spouse's interpersonal withdrawal. When paramedic experience higher stress and/or burnout at work, it increases spouse' interpersonal withdrawal at home. Spouses' enhanced withdrawal is also associated with increased marital tension. |
| Lorenzo & Carrisi (2020) [25] | Italy | Prospective observational study | HCWs (Sub-groups not stated) | HCWs (*n* = 38), their family members (*n* = 81) | Examining the risk of COVID-19 transmission from HCWs to their family members | • Lower infection risk was found for HCWs compared to their family members.<br>• HCWs were not a main source for the transmission to their family members. |
| Sachdeva et al., (2022) [27] | India | Cross-Sectional | HCWs (Medical and Paramedical roles) | Family members (*n* = 150) | Identifying the perceived stress, resilience and coping tendencies of family members of HCWs who had been working on COVID-19 during the pandemic in India | • High level of perceived anxiety and depression symptoms for family members.<br>• Majority of the family members reported low resilience and coping scores |
| Tüğen et al., (2023) [30] | Turkey | Cross-sectional | HCWs (Doctors, Nurses, Dentists, and Others not specified) | HCW (*n* = 144) and their children (*n* = 135) | Examining the anxiety and associated factors of HCWs and their children during the first wave of the pandemic in Turkey. | • Children whose HCW parents worked directly with COVID-19 patients reported significantly higher SCARED scores compared to children whose HCW parent did not worked with COVID-19 patients directly. |
| Ying et al., (2020) [31] | China | Cross-sectional | HCWs (Sub-groups not stated) | Family members (*n* = 845) | Assessing the mental health of family members of HCWs during the COVID-19 pandemic. | • Family members reported that they experience anxiety and depression.<br>• Family members of HCWs who work longer hours and work closely with the COVID-19 patients tended to experience higher anxiety and depression. |

nine were qualitative [36–44]. Of the 20 papers, five studies were based on participants in North America (USA and Canada) [32,34,36,37,43] eight in Asia (Iran, Hong Kong, China, and India) [26–29,31,33,35,38], five in Europe (UK, Sweden, Italy, Turkey, and France) [25,30,40–42], and two in Australia and New Zealand [39,44]. Fifteen studies focused on the experiences and mental health issues of families of HCWs during the COVID-19 pandemic. Five studies focused on the experiences of families of HCWs regardless of the COVID-19 pandemic [34,36,37,42,44]. Three studies directly focused on nurses [28,35,41], two on doctors [26,44], and four on EMS workers such as paramedics and ambulance service workers [32,34,36,37]. Five studies were related to spouses and partners of HCWs, one

**Table 3. Characteristics of the included qualitative studies.**

| First Author (Year) | Location | Study Design | High-Risk Occupational Group | Study Population | Research question | Main Findings |
|---|---|---|---|---|---|---|
| Chandler-Jeanville et al., (2021) [41] | France | Semi-structured interview | Nurses | Nurses (*n* = 49) and families (*n* = 48) | Reporting the experiences of family members of nurses after the first wave of the COVID-19 in France | • Being family members of nurses during the first wave of the COVID-19 pandemic had negative impacts on families.<br>• They experienced intense fear anxiety because of the high infection risk for their HCW family members.<br>• They were overwhelmed by the information provided media which some of them were accurate but some of them were not. |
| Emmett et al., (2013) [44] | New Zealand | Semi-structured interview | Paediatricians | Spouses (*n* = 10) | Identifying the positive and negative effect of paediatric work on family members and spouses of the paediatricians | • Spouses' sacrifices such as while choosing the living location because of paediatrician family member's work,<br>• Challenges in communication<br>• Lack of quality time as a family because of the long working hours and "after hours on-call" |
| Ericson-Lidman & Strandberg (2010) [42] | Sweden | Semi-structured interview | HCWs (Nurses, Social workers, Occupational therapists, Physiotherapists) | Family members and friends (*n* = 5) | Investigation of the family members and friends of healthcare workers who experience burnout | • Having so many responsibilities as a family member of a frontline rescue worker such as domestic responsibilities, supporting the worker emotionally |
| Mohammadi et al., (2022) [38] | Iran | Semi-structured interview | HCWs (Sub-groups not stated) | Family members (*n* = 25) | Exploring the experiences of family members of HCWs during the COVID-19 pandemic in Iran | • Family members reported experiences for both psychological tension (indescribable fear and worry, longing to see their loved ones, patient stone, bitter farewell, fear of the future) and dignity (acclamation, appreciation, feeling proud, spiritual growth). |
| Regehr (2005) [36] | Canada | Interview | Paramedics | Spouses (*n* = 14) | Exploring the effect of trauma exposure on paramedics' spouses | • Worrying about the firefighters because of the risks of the job<br>• Negative impact of occupational trauma on family relationships<br>• Using humour as a positive coping strategy |
| Roth & Moore (2009) [37] | US | Semi-structured interview | Emergency Medical Service workers | Family members (*n* = 12) | Identifying the factor which may impact the family system of the emergency medical service families | • Negative impact of shift work on family social lives, drops from the joint social activities as family because of the shift works<br>• Changes in marital and parental roles and having more domestic responsibilities as a family member of emergency service workers<br>• Family members reported that they concern about their EMS worker family member's safety.<br>• Family members pointed out that they developed some strategies to cope their EMS worker family member's job stress such as seeking social support, thinking positive, negotiating family responsibilities |

*(Continued)*

**Table 3.** (Continued)

| First Author (Year) | Location | Study Design | High-Risk Occupational Group | Study Population | Research question | Main Findings |
|---|---|---|---|---|---|---|
| Schaffer et al., (2022) [43] | US | Semi-structured interview | HCWs (Nurse, Respiratory therapist, Doctor, Physician's assistant, X-ray technician) | HCWs (n = 28) | Examining the impact of COVID-19 pandemic on HCW's functionality and its perceived impact on family members | • Increased responsibilities at home for families<br>• Due to the transmission risk, new hygiene rules at home such as social distancing and isolation<br>• Stigma for family members of healthcare workers and society's attitude<br>• Psychological distress and concern's about children's mental health and wellbeing |
| Sheen et al., (2022) [39] | Australia | Interview | Frontline HCWs (Allied health professionals, Nurses, Doctors) | Frontline HCWs (n = 39) | Discovering the effect of COVID-19 pandemic on families of frontline HCWs in Australia | • Changed roles and increased responsibilities at the home for families<br>• Concerns about HCWs lives and family members' lives because of the risk of contamination<br>• Either spending more time as a family or having more personal time |
| Tekin et al., (2022) [40] | UK | Semi-structured interview | HCWs (Ambulance drivers, Doctors, Physiotherapists) | Family members and friends (n = 14) | Exploring the experiences, views, needs, and mental health issues of family members and close friends of HCWs who was working on COVID-19 during the pandemic in the UK | • There was an increased domestic responsibility for family members such as childcare and cleaning because of HCWs long working hours<br>• Family members were worried about HCW#s life and safety, but also, they were worried about whole families' health because of the high risk of contamination<br>• Family members felt pride about what HCWs did during the pandemic<br>• Families also stated that there was a lack of recognition by the rest of the society about families sacrifices. For example, some of them mentioned that they had to give up from some elements of their job, because their responsibilities at home increased.<br>• Potential vicarious trauma for family members |

was related to children, and fourteen studies focused on the families and close friends of HCWs together. The smallest sample size was five [42] and the largest sample size was 39 [39] amongst the included qualitative studies. The smallest sample size was 60 [32] and the largest sample size was 992 [33] amongst the included quantitative studies. All studies were published between 2005 and 2023. The data collection methods used included surveys (n = 10) [26–35] and interviews (n = 9) [36–44]. One study focused on the transmission risk of COVID-19 from HCWs to families which used blood tests to determine transmission risk for families [25].

**Table 4. Number of qualitative studies (*n* = 9) meeting CASP criteria.**

|  | Totally Met | Partially Met | Not Met |
|---|---|---|---|
| 1. Was there a clear statement of the aims of the research? | 9 | 0 | 0 |
| 2. Is a qualitative methodology appropriate? | 9 | 0 | 0 |
| 3. Was the research design appropriate to address the aims of the research? | 9 | 0 | 0 |
| 4. Was the recruitment strategy appropriate to the aims of the research? | 9 | 0 | 0 |
| 5. Was the data collected in a way that addressed the research issue? | 9 | 0 | 0 |
| 6. Has the relationship between researcher and participants been adequately considered? | 6 | 2 | 1 |
| 7. Have ethical issues been taken into consideration? | 6 | 2 | 1 |
| 8. Was the data analysis sufficiently rigorous? | 8 | 1 | 0 |
| 9. Is there a clear statement of findings? | 9 | 0 | 0 |
| 10. How valuable is the research? | 9 | 0 | 0 |

## 3.3. Quality appraisal

We assessed the quality of the qualitative studies using the CASP checklist for qualitative studies [18]. A three-point scale was used as recommended by Lachal, Revah-Levy, Orri and Moro [45] to categorise criteria as totally met, partially met, and not met. The results of the CASP checklist for qualitative studies are shown in Table 4 (Also, see Supporting Information 4 in S1 File).

The quantitative studies were all cross-sectional designs. To assess the quality of these studies, we therefore used the Appraisal tool for Cross-Sectional Studies (AXIS) [19]. The results of the AXIS for quantitative papers are shown in Table 5 (Also, see Supporting Information 5 in S1 File).

Overall, the ratings of the qualitative research were high. Similarly, overall ratings of quantitative research were good. However, none of the studies justified their sample sizes and some of the sample sizes stated were very small.

## 3.4. Narrative synthesis

Findings were synthesised by outcomes. A summary of the findings from quantitative studies including the measures that were used in the studies and the identified risk factors is shown in Table 6. The qualitative findings are then briefly summarised with example quotes. The quantitative and qualitative findings are then narratively synthesised, exploring patterns across the included studies.

A total of 17 different scales were used to understand the impact of occupational stress on family members of HCWs in the included quantitative studies. Psychological distress amongst family members was assessed using Kessler's Psychological Distress Scale (K10) and the Perceived Stress Scale. The K10 [46] and the Perceived Stress Scale [47] both include ten items to examine the degree of psychological distress experienced by individuals in the last four weeks.

To assess anxiety and depression, authors used the Hospital Anxiety and Depression Scale (HADS), Generalised Anxiety Disorder-7 (GAD-7), the Screen for Child Anxiety-Related Emotional Disorders (SCARED), and the Patient Health Questionnaire-9 (PHQ-9). The HADS includes fourteen items to measure individuals" anxiety and depression symptoms in the past week [49]. The GAD-7 is a self-report questionnaire to examine anxiety symptoms in individuals in the last two weeks [52]. SCARED has 41 questions and five subscales (somatic and panic, generalised anxiety, social anxiety, separation anxiety, and school anxiety) to assess

**Table 5. Number of quantitative studies (*n* = 11) meeting AXIS criteria.**

| | Yes | Don't Know | No |
|---|---|---|---|
| **Introduction** | | | |
| 1. Were the aims/objectives of the study clear? | 11 | 0 | 0 |
| **Method** | | | |
| 2. Was the study design appropriate for the stated aim(s)? | 11 | 0 | 0 |
| 3. Was the sample size justified? | 0 | 0 | 11 |
| 4. Was the target/reference population clearly defined? (Is it clear who the research was about?) | 11 | 0 | 0 |
| 5. Was the sample frame taken from an appropriate population base so that it closely represented the target/reference population under investigation? | 9 | 0 | 2 |
| 6. Was the selection process likely to select subjects/participants that were representative of the target/reference population under investigation? | 8 | 0 | 3 |
| 7. Were measures undertaken to address and categorise non-responders? | 5 | 0 | 6 |
| 8. Were the risk factor and outcome variables measured appropriate to the aims of the study? | 10 | 1 | 0 |
| 9. Were the risk factor and outcome variables measured correctly using instruments/measurements that had been trialled, piloted or published previously? | 11 | 0 | 0 |
| 10. Is it clear what was used to determined statistical significance and/or precision estimates? (e.g. p-values, confidence intervals) | 10 | 0 | 1 |
| 11. Were the methods (including statistical methods) sufficiently described to enable them to be repeated? | 11 | 0 | 0 |
| **Results** | | | |
| 12. Were the basic data adequately described? | 10 | 0 | 1 |
| 13. Does the response rate raise concerns about non-response bias? | 7 | 2 | 2 |
| 14. If appropriate, was information about non-responders described? | 3 | 0 | 8 |
| 15. Were the results internally consistent? | 10 | 1 | 0 |
| 16. Were the results presented for all the analyses described in the methods? | 11 | 0 | 0 |
| **Discussion** | | | |
| 17. Were the authors' discussions and conclusions justified by the results? | 11 | 0 | 0 |
| 18. Were the limitations of the study discussed? | 7 | 0 | 4 |
| **Other** | | | |
| 19. Were there any funding sources or conflicts of interest that may affect the authors' interpretation of the results? | 0 | 0 | 11 |
| 20. Was ethical approval or consent of participants obtained? | 9 | 1 | 1 |

anxiety symptoms in children [53]. The PHQ-9 aims to examine the depressive symptoms of individuals over the preceding two weeks with nine questions [50].

Regarding the PTSD symptoms, burnout, and rumination, the Self- Rating Inventory for PTSD (SRIP), the Maslach Burnout Inventory–Human Services Survey (MBI-HSS), and the Rumination-Reflection Questionnaire (RRQ) were used in the included studies. The SRIP includes 22 items to examine the severity of the PTSD with items based on DSM-IV criteria for PTSD [54]. The MBI-HSS includes 22 items and three subscales to assess the daily burnout experiences of individuals [58]. The RRQ includes 15 items that measure ruminative self-focus in individuals such as constantly thinking about how an individual acted in a previous event [59].

In terms of assessing coping skills and resilience, authors used the Brief Resilience Coping Scale (BRCS), 10-items Connor-Davidson Resilience Scale, The Social Support Questionnaire (SSQ), the Brief Ways of Coping Inventory, and the Coping Responses Inventory (CRI). The

**Table 6. Detailed findings from quantitative studies of the occupational stress for family members of healthcare workers.**

| Type of outcomes studied | Study and prevalence | Measures used | Risk factors identified |
|---|---|---|---|
| **1. Mental Health Outcomes** | | | |
| Worry | *Goud et al., (2021)* [26]<br>In this cross-sectional study of 120 spouses of doctors in India, 72.5% of the spouses of doctors who worked during the COVID-19 pandemic reported psychological distress | Kessler's Psychological Distress Scale (K10) [46] | Financial insecurity |
| | *Sachdeva et al., (2022)* [27]<br>In this cross-sectional survey of 150 family members of HCWs in India, 75% of the family members of HCWs reported moderate to high levels of perceived stress during the COVID-19 pandemic. | • Perceived Stress Scale [47],<br>• Brief Resilience Coping Scale [48],<br>• Hospital Anxiety<br>• Depression Scale [49] | Being a female family member |
| Anxiety and depression | *Banitalebi et al., (2021)* [28]<br>22.6% of family members of nurses reported mild depression, 71.4% of families reported moderate depression, and 1.8% of them reported severe depression | Patient Health Questionnaire-9 (PHQ-9) [50] | Families of nurses found that mental health problems were more common in family members who are older than 57 years old, that there was a statistically significant relationship between being female and experiencing mental health problems, and that married partners tended to experience more mental health problems than unmarried partners. |
| | *Feng et al., (2020)* [29]<br>In this cross-sectional survey study of 671 family members of frontline worker in China, 49% of the family members of frontline workers reported mild and above anxiety symptoms, 12.2% of the family members had clinically significant depression symptoms | • Perceived Stress Scale [47],<br>• 10-items Connor-Davidson Resilience Scale [51],<br>• Generalised Anxiety Disorder-7 (GAD-7) [52],<br>• Patient Health Questionnaire-9 PHQ [50] | Families experienced high anxiety due to the uncertainty of the situation and lack of knowledge about whether their frontline worker family members would return home. |
| | *Sachdeva et al., (2022)* [27]<br>23% and 17% of family members of HCWs in India experienced clinically significant anxiety and depressive symptoms, respectively during the COVID-19 pandemic. | • Perceived Stress Scale [47],<br>• Brief Resilience Coping Scale [48],<br>• Hospital Anxiety and Depression Scale [49] | It was reported that female family members tend to experience higher level of anxiety and depression compared to male family members. Family members who are younger than 40 years old, have a child or have an elderly relative at home (Sachdeva et al., 2022) [27], and work in private sectors compared to government or institutional employees (Ying et al., 2020) [31] were at risk of experiencing higher anxiety and depression symptoms.<br>Family members who are younger than 40 years old, have a child or have an elderly relative at home were at risk of experiencing higher anxiety and depression symptoms.<br>Furthermore, families of HCWs whose HCW family member worked directly on COVID wards, worked more than 48 hours per week, and worked in poor conditions such as having a lack of protection equipment, tended to experience higher levels of anxiety and depression. |
| | *Tüğen et al., (2023)* [30]<br>In this cross-sectional study which was conducted with 145 HCWs and their 135 children in Turkey, children's mean SCARED subscale scores were: for panic/somatic: 7.23 ± 5.71; for general anxiety: 6.89 ± 4.25; for separation anxiety: 6.88 ± 3.94. | Screen for Child Anxiety-Related Emotional Disorders (SCARED) [53] | Findings of a cross-sectional study conducted with HCWs and their children whose ages were between 8 to 18, showed that children had significantly higher somatic/panic subscale scores, generalised anxiety subscale scores, and separation anxiety scores when their HCW parent was directly involved in the care of COVID-19 patients. |
| | *Ying et al., (2020)* [31]<br>This cross-sectional study was conducted with 845 family members of HCWs in China, | • Generalised Anxiety Disorder-7 (GAD-7) [52]<br>• Patient Health Questionnaire-9 (PHQ-9) [50] | Female family members tend to experience significantly higher anxiety symptoms compared to male.<br>Family members who work in private sectors compared to government or institutional employees.<br>Family members whose HCW family member worked directly on COVID wards, worked more than 48 hours per week, and worked in poor conditions such as having a lack of protection equipment. |

*(Continued)*

**Table 6.** (Continued)

| Type of outcomes studied | Study and prevalence | Measures used | Risk factors identified |
|---|---|---|---|
| Secondary Traumatic Stress and PTSD | **Feng et al., 2020** [29]<br>In a cross-sectional study which included 671 family members of first responders during the pandemic, researchers reported that 10.4% of family members may experience PTSD symptoms. | • Perceived Stress Scale [47],<br>• 10-items Connor-Davidson Resilience Scale [51],<br>• Generalised Anxiety Disorder-7 (GAD-7) [52],<br>• Patient Health Questionnaire-9 PHQ [50] | No risk factors identified |
| | **Henry et al., (2023)** [32]<br>HCWs' mental health has an impact on their loved one's mental health. According to the findings of this cross-sectional study, which was conducted with 30 couples in the US, partners of emergency service workers (emergency service technicians or paramedics) are at a higher risk of experiencing PTSD symptoms when the emergency service worker is diagnosed with PTSD. | • The Self- Rating Inventory for PTSD (SRIP) [54],<br>• The Couple Satisfaction Index [55],<br>• The Social Support Questionnaire [56] | HCW's PTSD |
| Emotional Burden | There was no quantitative study that focused on social support. | | |
| **2. Family Relationships** | | | |
| Family functioning and relationships | **Chua et al., (2021)** [33]<br>There was a positive relationship between higher perceived stress in HCWs and more negative changes in family relationships. | • Perceived Stress Scale (PSS-10) -Perceived Stress Scale [47],<br>• Family APGAR (Adaption, Partnership, Growth, Affection, Resolve) Scale [57], | No risk factors identified |
| Couple relationships | **Henry et al., (2023)** [32]<br>In this cross-sectional study with 30 couples in the US, PTSD symptoms of emergency medical service workers do not have a significant effect on the relationship satisfaction that was reported by their partners. | • The Self- Rating Inventory for PTSD (SRIP) [54],<br>• The Couple Satisfaction Index [55],<br>• The Social Support Questionnaire [56] | No risk factors identified |
| | **King et al., (2014)** [34]<br>In this study which was conducted with 87 couples (paramedics and their spouses') in Canada, enhanced withdrawal was associated with increased marital tension over time. | • Perceived Stress Scale [47],<br>• Maslach Burnout Inventory–Human Services Survey (MBI-HSS) [58]<br>• Rumination-Reflection Questionnaire [59],<br>• Brief Ways of Coping Inventory [60] | No risk factors identified |
| Absence and Separation | There was no quantitative study focused on the absence and separation. | | |
| **3. Coping Skills and Resilience** | | | |
| Coping Skills | **Banitalebi et al., (2022)** [35]<br>Coping skills had an important direct impact on psychological health and quality of life amongst family members of HCWs. | • Patient Health Questionnaire-9 (PHQ-9) [50]<br>• Coping Responses Inventory [61], | No risk factors identified |
| | **Sachdeva et al., (2022)** [27]<br>More than 50% of the family members had low resilience and coping scores. | • Perceived Stress Scale [47],<br>• Brief Resilience Coping Scale [48],<br>• Hospital Anxiety and Depression Scale [49] | No risk factors identified |
| Social Support | There was no quantitative study that focused on social support. | | |

*(Continued)*

**Table 6.** (Continued)

| Type of outcomes studied | Study and prevalence | Measures used | Risk factors identified |
|---|---|---|---|
| **4. Quality of Life and Social Life** | | | |
| Life Satisfaction | ***Banitalebi et al., (2022)*** [35]<br>30.77% of family members of nurses reported poor quality of life, 27.88% reported moderate quality of life, and %41.35 reported good quality of life. In the same study, researchers also examined the different subscales of the quality of life for family members, and they found that while physical functioning had a maximum mean, social functioning had a minimum mean score. | • Patient Health Questionnaire-9 (PHQ-9) [50]<br>• Coping Responses Inventory [61], | No risk factors identified |
| Social Life | There was no quantitative study that focused on social life. | | |
| **5. Practical Outcomes** | | | |
| Domestic responsibilities (cleaning, paying the bills, taking care of vulnerable relatives, childcare, shopping, organising family vacations and activities) | ***Feng et al., (2020)*** [29]<br>40.2% of the family members reported that their daily life was significantly impacted due to their support for frontline workers. | • Perceived Stress Scale [47],<br>• 10-items Connor-Davidson Resilience Scale [51],<br>• Generalised Anxiety Disorder-7 (GAD-7) [52],<br>• Patient Health Questionnaire-9 PHQ [50] | No risk factors identified |
| Choosing living location | There was no quantitative study that focused on choosing a living location. | | |

BRCS aims to investigate how individuals cope with a stressor using four questions [48]. Connor and Davidson (2003) define resilience as growth in the face of challenges measured in their 10-items Connor-Davidson Resilience Scale. [51]. The SSQ includes 27-item to examine the social support resources of individuals and how satisfied individuals are with the social support they receive [56]. The Brief Ways of Coping Inventory was designed based on the big-five traits (Neuroticism, Extraversion, Openness to Experience, Agreeableness, and Conscientiousness) to evaluate how individuals cope with stress [60]. The quantitative study included in this review only used two items from this scale "(a) withdrew from the other person(s) involved, (b) gave the other person(s) involved the 'silent treatment,' and (c) sulked" (s King et al., (2014), p. 463). The CRI was developed to examine the coping responses of individuals by using 32 items [61].

Authors assessed couple satisfaction and family relationships by using The Couple Satisfaction Index and Family APGAR (Adaption, Partnership, Growth, Affection, Resolve) Scale. The Couple Satisfaction Index includes 32 items to measure the relationship between couples and how satisfied they are in their romantic relationship [55]. The Family APGAR Scale was designed to assess the family systems regarding adaptation, partnership, growth, affect, and resolve in the family, and includes five questions [57].

Authors of the included studies reported that the questionnaires that they used in their studies had good reliability and validity.

**3.4.1. Mental health outcomes.** *Worry*. Eight studies explored the potential worry experienced by families of HCWs. Only two studies explored this prior to COVID and reported that spouses of paramedics [36] and families of EMS [37] experienced high levels of stress due to concerns about physical safety, working conditions (unhealthy foods in the canteen, long working hours) and safety risks to their HCW family member at work.

Six studies explored worry in families of healthcare workers in the context of COVID-19 specifically [25–27,38–40]. During the COVID-19 pandemic, family members were also worried about the physical safety of their HCW loved ones. For example, in a qualitative study conducted with 25 family members of HCWs in Iran during the pandemic, family members whose HCW wife or daughter was pregnant, specifically worried about both their wife's/daughter's lives as well as the life of the unborn child [38]. One of the spouses of a HCW shared his feelings with Mohammadi et al., [38]: "*My wife is 24-week pregnant. She loves her job and says she became a doctor for times like this. I understand her, but I can't help worrying. I'm afraid of the future. What if something happens to her and puts her life in danger. I'm afraid of premature birth, having a premature baby, and many complications that may follow. We don't know how this unknown disease affects mothers and their babies. Thinking about the future and uncertainty about what it holds is always with me.*" Families of HCWs were particularly worried that HCW would bring the disease home and that their children and other families would also contract it [39,40]. In a prospective observational study, which was conducted with 38 HCWs and their 81 family members in Italy, infection rates were lower for HCWs compared to their families, and researchers pointed out that HCWs were not a main source for the transmission of the COVID-19 for their families [25]. However, even though HCWs may not be the main source of transmission, there was still concern amongst family members about transmission risks [40].

*Anxiety and depression*. Seven papers focused on anxiety and depression experienced by families and friends of HCWs [27–31,38,41,42].

Only one study focused on the experiences of families before the COVID-19 pandemic. This qualitative interview study was conducted with five family members and close friends of HCWs with burnout in Sweden [42]. Family members reported that they were worried because they were struggling to understand HCW's burnout experiences. Additionally, due to

the burnout and job stress, family members were required to undertake more responsibilities in the home and family members described feeling anxious about how their daily lives were disrupted while they were taking more responsibilities at the home. However, it is important to highlight that this study failed to provide any detail about the demographics of participants, so it is difficult to know how transferable the findings of this study might be.

Six papers identified anxiety and depression in the families of HCWs in the context of the COVID pandemic. Families of frontline workers who had been working during the COVID-19 pandemic [29], stated that they experienced high anxiety. Similarly, families of HCWs in Iran [28,38], France [41], China [29,31], India [27], and Turkey [30], experienced intense anxiety and depression during the pandemic. Families mostly tended to be concerned about HCW's health and working conditions [29,40], for example, having enough grocery supplies and when family members would be able to see their HCW loved one [29]. However, since these studies were conducted at a single point in time during the COVID-19 pandemic and there is no data on the mental health of the participants before COVID-19, these results should be considered carefully.

*Secondary traumatic stress and PTSD.* Four studies reported experiences of secondary traumatic stress and PTSD in families of HCWs, and all of these were conducted during or after the COVID-19 pandemic [29,32,38,40]. Families of HCWs who were working during the COVID-19 pandemic reported that they had vivid dreams about the traumatic situations that happened at the HCW's work [40]. Likewise, it was reported that HCWs were sharing traumatic work experiences with families to seek support, but this may increase the risk of experiencing secondary traumatic stress amongst family members [38,40]. However, these studies' sample sizes are very small making it hard to draw robust conclusions. Additionally, all four studies were either conducted during or after the COVID-19 pandemic, with no comparisons about the mental health and wellbeing of those families and HCWs before the pandemic.

*Emotional burden.* There were four studies that reported on the emotional burden that families experienced. Families of HCWs tended to see themselves as a source of support for the HCW and made emotional sacrifices, both before and during the pandemic.

Two studies were conducted before the pandemic and researchers reported that families tended to carry the emotional burden, protecting the rest of the family from the details of traumatic events that their HCW loved one experienced, "walking on tiptoe" [42], and trying to read the emotions of the worker and the level of the worker's exhaustion, the expression on the worker's face, or the lack of communication and try to comfort them [36].

Two studies were conducted during the pandemic and in these studies family members reported that they just tried to listen their HCW family member [40] and hide their own anxiety and fear to support them [38]. These findings show that family members experience emotional burden while supporting their HCW family member, however, the COVID-19 pandemic may have aggravated this.

**3.4.2. Family relationships.** *Family relationships and functioning.* Seven studies investigated the relationship between work stress and its impact on family relationships. Results across these studies were consistent, showing that HCWs' stress had a negative impact on family relationships before and during the COVID-19 pandemic. Families of HCWs in Hong Kong [33], and spouses of paramedics in Canada [36] demonstrated that higher stress experienced by the HCWs was correlated with more negative family relationships. In a qualitative study which included 14 spouses of paramedics, spouses stated that there was an extreme negative impact of the paramedic's stress and trauma on family relationships. A husband shared his experiences: "*She crowds in on herself. She becomes very quiet, won't talk. And of course, the flip side of that is if you press the wrong button, then BOOM!*" [36]. Some studies conducted during

the COVID-19 pandemic have shown that conflicts experienced by families of HCWs in their family relationships have increased. For instance, in a study of 39 frontline HCWs, participants reported that they started to spend more time with their families, but this was emotionally draining because both families and managers required more time from HCWs and this caused conflicts at home [39]. Similarly, in a qualitative study, 28 HCWs pointed out that there was an enhanced tension in family relationships due to the financial concerns caused by COVID-19, one-sided parental decisions, or decreased couple's time. For example, a nurse stated conflicts in family relationships due to parenting decisions: "*I guess I am more restrictive with the kids and what they can do. I would rather have them not do some things and go some places. My husband is less weary than I am about it so that can create some tension*" [43].

There were, however, studies which focused on the improvements in family relationships during the pandemic in HCW families. For example, according to the findings of a qualitative study which was conducted with 49 nurses and 48 family members in France, nurses and their family's perceived lockdown as an opportunity to build stronger relationships with family members and to spend more time together cooking, baking, and playing games [41]. Similarly, Schaffer et al., [43] highlighted that there was a stronger bond between family members, and they were willing to help each other compared to before pandemic. However, these study participants were mostly female. More research is required with male family members.

Four studies reported a sense of pride amongst families of HCWs. For example, family members of HCWs who had been working during the COVID-19 pandemic in the UK [40], France [41], and Iran [38], reported that despite the lack of adequate equipment at the beginning of the epidemic, the high risk of contracting the disease, and poor working conditions, HCWs continued to save lives, and this led to a great sense of pride for families. Even before the COVID-19 pandemic, spouses of paramedics in Canada reported being proud of their HCW family members [36].

*Couple relationships.* Six studies focused on the relationship between occupational stress and couple relationships and intimacy, reporting that job stress may have a negative impact on couple relationships and intimacy both before and during the pandemic [34,36,38,40,44]. For example, before the pandemic, spouses of paediatricians reported that because of job stress and long working hours, they experienced intimacy and communication challenges [44]. Similarly, a longitudinal study's findings highlighted that there was a relationship between paramedics' perceived stress and burnout experiences at work and spouses' interpersonal withdrawal [36]. Studies which were conducted during the pandemic also support these findings. For example, spouses of HCWs in the UK stated that their sacrifices were not recognised by their HCW partners and society [40]. Additionally, lack of privacy during the pandemic also caused some tension in couple relationships: "*I think there is some tension in the marriage because the kids are around more. My husband and I are not getting as much alone time together and individually because the kids are around.*" [43]. However, this study's sample size is small.

*Absence and Separation.* Four studies reported the negative impact of HCWs being absent and separated from their families and they were all conducted during the pandemic.

HCWs who worried that they might spread the virus to their families often isolated themselves from their loved ones. For example, in a qualitative study in the UK, it was reported that because of the long working hours and shifts, HCWs tended to be away from their families: "*Our kids didn't get to see as much of their dad, and they missed him as well.*" [40]. Similarly, a study conducted with 25 family members in Iran reported that HCWs could not come back to their home regularly because of long working hours and shifts which significantly impacted family members, especially children [38].

Even when HCWs were at home with their families, there was still often separation. For instance, Chandler-Jeanville et al., [41] reported that, due to transmission risk, some nurses

limited their physical contact with their families. This was especially challenging for children who wanted to hug and kiss the HCW family member, but spouses and partners also stated that this limited physical contact impacted their relationship negatively as well. Schaffer et al., [43] supported these findings in their research which was conducted with 28 HCWs in the US. They reported that because of limited physical contact, families started to be creative in terms of communicating with the HCW. A nurse shared her experiences: "*The girls text and facetime me more from their rooms in the house, which I used to never let them do.*" This helped them to build new routines and rituals to retain their relationships.

**3.4.3. Coping skills and resilience.** *Coping skills*. Seven studies focused on the impact of coping skills on psychological health and quality of life. The results of the studies were consistent, providing important knowledge about coping skills both before and during the COVID-19 pandemic. Studies conducted before the pandemic pointed out that family members and friends of HCWs in Sweden stated that searching for recuperation and learning something new about themselves helped them to re-energise and find strength to cope with the healthcare work stress [42]. Similarly, twelve family members of EMS workers in the US pointed out that developing their own interests helped them to cope with the impact of EMS work [37]. Additionally, emotional support, positive thinking, and sharing domestic responsibilities were helpful for families to cope with the HCW's job [37]. Similarly, studies conducted during the pandemic reported that coping skills had an important direct impact on psychological health and quality of life amongst families of HCWs [27,28,35].

One study focused on humour as a coping strategy against occupational stress. In this study which was conducted with 14 spouses of paramedics in Canada, spouses reported that they used humour with their HCW spouses to reduce the impact of tragic events. *"We've developed a very left field sense of humour. It's questionable, but it's good"* [36]. However, this study focused on the experiences of families of HCWs before the COVID-19 pandemic. The perspective on humour and its use may have changed during the pandemic, when life was in serious danger not only for healthcare professionals but also for their families.

One study focused on religion as a coping strategy against occupational stress. During the pandemic, 22 out of 25 family members of HCWs in Iran reported that they spiritually grew during the pandemic and prayed for comfort and safety for everyone. One participant said that "*Since COVID-19 began to spread; I have done more talking with God, vows, good deeds, and altruism. I feel more spiritual than before*" [38]. However, since there is no other research which focused on the spirituality of families of HCWs, it is difficult to generalise these findings.

*Social support*. Six studies explored the impact of social support on coping with occupational stress amongst family members, with consistent findings pointing out the importance of social support to cope with occupational stress.

Two studies focused on social support before the pandemic and researchers reported that, thanks to social support, families of EMS workers in the US coped with the HCW's job stress [37]. Spouses of paramedics pointed out another important topic: paramedics mostly had peer support during their shift, but that was not enough [36].

Four studies focused on the families' experiences of social support during the pandemic. In their qualitative study which was conducted with nurses and their families in France, Chandler-Jeanville et al., [41], reported that families were sincerely grateful to their friends and extended family members for their support during the pandemic. Additionally, they were happy to hear handclaps and to receive presents from the local community because they tended to interpret them as evidence of social support [41]. Similarly, in a qualitative study which included 25 family members in Iran, 23 of them pointed out that they felt social support by the rest of society showed their gratitude to families of HCWs [38]. On the other hand,

families of HCWs were worried that this appreciation would fade away too quickly and HCWs' working conditions would not be improved [40,41]. In another qualitative study conducted with 28 HCWs in the US, HCWs reported that their families were stigmatised because of their healthcare work. For instance, one nurse manager shared her experience: "*I stopped telling people that I was a nurse in public. I told my kids to stop telling people that I was a nurse because people were afraid of me because of potential exposure to COVID-19*" [43].

**3.4.4. Quality of life and social life.**    *Life satisfaction.* Two studies explored life satisfaction amongst family members of HCWs. Families of HCWs who had been working on COVID-19 during the pandemic in the UK stated that they had to sacrifice many elements of their own work because of increased shifts of the HCW family member, and this impacted their job satisfaction [40]. Additionally, in a cross-sectional study conducted with 220 family members of nurses in Iran, researchers found that 30.77% of family members reported poor quality of life [35]. However, these studies were conducted during the COVID-19 pandemic. We do not have information about quality-of-life experiences of families of HCWs before the pandemic.

**Social life.**    Four studies examined the impact of shift work and long working hours on the social life of families. We found that shift work had a significant negative impact on the social life of families of HCWs and their experiences were similar before and during the pandemic.

Two studies were conducted before the COVID-19 pandemic. Findings of a qualitative study in the US with families of emergency medical workers demonstrated that shift work has a negative impact on family social life [37]. Likewise, for some HCWs there were difficulties in keeping their work/life balance due to shifts and long working hours. They reported that even if families can spend more time together despite shift work and long working hours, there will be some costs. Spouses of paediatricians [44] in New Zealand stated that while they spend time with their families, they do not have time for activities as a couple.

Two studies focused on the social life of families during the pandemic. In a qualitative study which was conducted with 39 frontline workers in Australia, HCWs tried to spend time with their families, but they were already working long hours. For this reason, spending time with their families came at the cost of losing personal space and "Me Time" [39]. Also, the social lives of families of HCWs were disrupted not only because of the HCW's long working hours and shifts, but also social isolation and stigma. Twenty-eight HCWs in the US reported that because of the infection risk, their family members were stigmatised and had to withdraw from social activities. They specifically reported that they were worried about the impact of stigma and social isolation on their children's mental health and wellbeing [43].

**3.4.5. Practical Outcomes.**    *Domestic responsibilities.* According to six studies with consistent results, family members of HCWs tended to take on more responsibilities at home, regardless of the pandemic. Families of HCWs stated that they have to be responsible for a lot of the domestic responsibilities that couples normally share because of the HCW's job demands. These responsibilities included cleaning, paying the bills, shopping, childcare, and supporting vulnerable family members [29,37,39,40,42]. According to findings of two studies focused on the experiences of families before the pandemic, family members tended to have more responsibilities for cleaning and childcare [37,42] with family members perceiving that if they take over domestic responsibilities from the HCW family member, they may recover from their job stress quicker [37]. Similarly, during the COVID-19 pandemic, family members were willing to take on more responsibilities at home to help the HCW, [39,40].

During the pandemic, however, family members' domestic responsibilities were increased not only because of the increased working hours and shifts, but also, because of the high-risk of carrying the disease home, family members tended to clean the house more than usual. A male partner of a physiotherapist who worked closely with COVID-19 patients during the pandemic in the UK reported that: "*I've helped out making a packed lunch and when she came*

*home from work every day, we got into a sort of routine where I would close all the curtains so she could strip off in front of the washing machine, put [her clothes] in the washing machine, and shower upstairs. So, I was helping out in that way*" [40]. Similarly, a HCW who worked with COVID-19 patients during the pandemic in the US said that: "*[My] husband goes around when I get home and wipes down and bleaches everything that I touch*" [43].

*Impact on living location*. Two studies focused on how the families of HCWs are also impacted by a lack of choice of living location. For example, ten spouses of paediatricians in New Zealand before the pandemic [44] and 14 family members of HCWs in the UK during the pandemic [40] pointed out that they have to choose their home's location based on the HCW, because of long working hours and shifts. Because of that choice, families of the HCWs sometimes needed to travel for several hours every day to go to their own job, which caused tension between family members. Results show that moving constantly due to HCW's work location has a negative impact on families regardless of the COVID-19 pandemic [40,44].

## 4. Discussion

In this review our aim was to understand the impact of occupational stress on family members of HCWs and how this impact varied before and during the COVID-19 pandemic. Based on the narrative synthesis of 20 studies, we identified five main outcomes for family members of HCWs.

Family members' experiences of many issues were similar before and during the COVID-19 pandemic. Firstly, many of the families of HCWs experienced mental health issues such as worry, depression, anxiety, and secondary traumatic stress both pre- and during the pandemic. Secondly, regardless of the pandemic, almost all family members in the included studies reported that occupational stress experienced by HCWs caused conflict in family relationships, and poorer functioning in the family. Long working hours and shift work could also negatively impact families in terms of social life and quality of life. Finally, family members of HCWs identified that because of the high demands of the healthcare work, family members tended to take on more responsibilities at home such as childcare, caring for vulnerable family members, paying the bills, and cleaning. According to the results of this review, emotional support, social support, positive thinking, humour, and religion helped family members to cope with the HCW's job stress and its potentially negative impact on their families.

There were also some different experiences of families of HCWs during the COVID-19 pandemic compared to before the pandemic. For instance, researchers reported that family members of HCWs tended to experience Secondary Traumatic Stress and PTSD symptoms. Additionally, during the pandemic, HCWs stayed away from home for longer periods of time due to long working hours, additional shifts, and the risk of transmission of the disease. This separation and absence from home caused distress to families. With increased working hours and additional shifts during the COVID-19 pandemic, family members often had to sacrifice their own jobs, which decreased their life satisfaction.

COVID-19 also worsened some experiences for family members. Firstly, the COVID-19 pandemic could exacerbate conflict in some healthcare families. Secondly, families reported even lower quality of social life due to the stigma attached to HCWs' families–that is, the rest of the society could view HCWs' families as a potential COVID-19 transmitter. Thirdly, families of HCWs tended to take on even more domestic responsibilities and cleaning during COVID-19. Finally, emotional burden may have been increased even more as family members tended to supress their emotions to help the HCW.

We identified potential relationships between some themes in the findings of this review. In terms of mental health and wellbeing, increased working hours of HCWs was associated with

increased mental health issues for families. Ying et al., [31] and Tugen et al., [30] reported that when HCWs spent more time with COVID-19 patients, family members tended to interpret this situation as an increased risk for HCW's life and they tended to experience higher anxiety and depression symptoms. Additionally, joint activities and spending time as a family may increase life satisfaction and decrease mental health issues [62]. However, due to long working hours and shifts, family activities and routines of HCW families were disrupted, and this may increase the mental health issues across family members of HCWs.

In previous literature, it has been well-documented that families of other high-risk workers such as police officers, firefighters, and military personnel are at risk of developing mental health issues. There are similarities between families of other high-risk workers and families of HCWs. For example, in a systematic review which focused on the families of emergency responders (police officers and firefighters), researchers reported a negative impact of life threats for high-risk workers and increased domestic responsibilities for families on family members' mental health and well-being [12]. Also, families of military personnel tend to experience worry, anxiety, and depression due to the absence of military personnel from home [63]. Our findings are consistent with this. Similarly, spouses of firefighters who were first responders after the World Trade Centre (WTC) attack [64] stated that when the firefighters left home to save the lives of others, they experienced high anxiety due to the uncertainty of the situation and lack of knowledge about whether they would return home. Ultimately, both other high-risk worker families and family members of HCWs appear to experience mental health issues and decreased well-being due to the uncertain and unsafe job environment of the workers, the workers absence from home, and increased domestic responsibilities for families.

Based on the findings of the included studies conducted in different countries, it may be that the experiences of families of HCWs vary depending on the culture they live in. Hofstede [65] mentioned that Asian countries are mostly collectivist which means that individuals are interconnected with their families and society, and they tend to support each other as a community to heal [66]. In the included studies, families from Asian and Middle Eastern countries reported that they felt the appreciation and applauses, but also, they felt a sincere support from the rest of society [27,38]. In our review study, we found that families from western countries reported that they also appreciated society's applauses and appreciation, but they worried that this will fade away too quickly. Also, some of the HCWs in western countries reported that they did not receive social support, and also felt stigmatised and seen as a transmitter of the disease by society [43]. In terms of coping, Taylor et al., [67] reported that individuals from different countries may use different coping strategies because they may tend to interpret the potential stressors differently. In our review, studies conducted in Asian countries reported on the importance of social support and family relationships. In addition to those, studies conducted in Western countries reported on the importance of couple relationship and individual coping strategies such as developing new interests and hobbies.

The primary findings of this review show that there is a potential risk to the mental health and well-being of families of HCWs. Very few papers looked at potential benefits or positive outcomes for families. Some of the family members of HCWs who had been working during the COVID-19 pandemic in the UK reported that they had a great sense of pride about the HCW's job [40], and some of the family members reported that their family relationships improved during the pandemic [43]. The potential positive impact of being a family member of a HCW remains a current gap in the literature.

### 4.1. Strengths and limitations

**4.1.1. Strengths and limitations of the included papers.** Most of the studies included in this review met the criteria for high-quality research. Yet, there are a number of limitations in the articles included in this review. Firstly, we aimed to include studies that focused on the experiences, views, needs and mental health issues of a variety of family members of HCWs. However, most studies focused on spouses, partners, and wives in heterosexual relationships, and children and teenagers of HCWs. This review found a gap in the literature, with a lack of research that focuses on the partners and spouses in same-sex relationships, parents, and siblings of HCWs. Secondly, most of participants in the included studies were female and there was a lack of research on male family members. Thirdly, most of the included studies reported on the mental health and wellbeing of family members during the COVID pandemic. There was no information in most studies about the previous mental health status of family members. Finally, in some of the qualitative studies included in this review, reflexivity was not included in the paper. For this reason, it is difficult to determine how the characteristics of the researchers who conducted this study may have impacted the data collection and analysis.

**4.1.2. Strengths and limitations of the systematic review.** In this review we have synthesized the results of qualitative and quantitative studies, according to the highest quality standards. We included studies from thirteen different countries from four continents. For this reason, our results are potentially transferrable to different countries and cultures. Our research team was diverse, including researchers from different career stages, clinical experiences, and different cultural groups. This allowed us to consider our findings from a variety of perspectives and build a rich and in-depth analysis. Yet, there are some limitations. The search was restricted to the English and Turkish languages due to the spoken languages of the researchers. Therefore, there may have been studies that were written in other languages that were missed.

### 4.2. Future research and implications

More research needs to be conducted regarding the experiences, needs, mental health, and well-being of families of HCWs. In the current published literature, the focus was mostly on the mental health of spouses, partners, and wives, and there is a significant gap in the literature regarding the experiences of the other family members and close friends of HCWs and the experiences of the spouses and partners from same-sex relationships. Therefore, it would be important in future research to explore the experiences of different family members and close friends, and in addition partners from same-sex relationships. There is a prominent gap about any positive impacts or potential benefits for healthcare workers' families, which could usefully be explored further. Additionally, there are limited studies which focus on vicarious and secondary trauma, and those that do, mostly concern the COVID-19 pandemic. Clinicians in occupational health and psychological health services need to be aware of, and trained to understand that families of HCWs are also at risk for mental health issues. Where possible, these clinicians could provide support to family members.

### 5. Conclusion

In this systematic review we aimed to understand the impact of occupational stress on families of HCWs before and during the COVD-19 pandemic. As a result of the narrative synthesis of 20 studies, we identified that there is a high risk for adverse mental health and well-being of HCWs' family members. HCWs are more at risk of experiencing mental health problems because of the nature of their jobs, and it can be challenging being the family member of someone with a mental health problem. Separately, because of the potentially traumatic nature of

healthcare work, family members may experience negative impacts on their own mental health by hearing about traumatic incidents, or they could be affected by the long hours, shift work, and compassion fatigue that their HCW family member experiences. This review shows the similar and different experiences, needs, and mental health issues of family members of HCWs before and during the pandemic. Organisations have legal, moral, and reputational responsibilities to protect HCWs and their families. In order to provide better support to family members, it is important to conduct further research to expand and address gaps identified in the literature, train the clinicians for clinical support, and extend the mental health services to family members. For instance, when workers engage with a service, clinicians should also consider the impact on and needs of their families. Additionally, it is necessary to increase organisational awareness of the impact of occupational stress on family members of HCWs.

## Supporting information

**S1 File.** Supporting Information file includes five supporting information documents which are described in the following:

- Supporting Information 1: Key Search Terms (including PsychINFO, Scopus, Medline, and Embase)

- Supporting Information 2: AMSTAR Checklist

- Supporting Information 3: PRISMA Checklist

- Supporting Information 4: CASP Results for Qualitative Studies

- Supporting Information 5: AXIS Results for Quantitative Studies
  (DOCX)

## Acknowledgments

We would like to thank the Ministry of Education in Turkey, who have supported Sahra Tekin for her PhD studies.

## Author Contributions

**Conceptualization:** Sahra Tekin, Naomi Glover, Jo Billings.

**Formal analysis:** Sahra Tekin.

**Investigation:** Jo Billings.

**Methodology:** Sahra Tekin, Helen Nicholls, Dannielle Lamb, Jo Billings.

**Software:** Sahra Tekin, Helen Nicholls.

**Supervision:** Naomi Glover, Jo Billings.

**Validation:** Sahra Tekin.

**Visualization:** Sahra Tekin, Dannielle Lamb.

**Writing – original draft:** Sahra Tekin.

**Writing – review & editing:** Sahra Tekin, Helen Nicholls, Dannielle Lamb, Naomi Glover, Jo Billings.

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
