## [Decision Letter · Decision Letter 0]

28 Jun 2024

PONE-D-24-09879Impact of Occupational Stress on Family Members of Healthcare Workers Before and During the COVID-19 Pandemic: A systematic reviewPLOS ONE

Dear Dr. Tekin,

Thank you for submitting your manuscript to PLOS ONE. After careful consideration, we feel that it has merit but does not fully meet PLOS ONE’s publication criteria as it currently stands. Therefore, we invite you to submit a revised version of the manuscript that addresses the points raised during the review process.==============================

We look forward to receiving your revised manuscript.

Kind regards,

Sirwan Khalid Ahmed

Academic Editor

PLOS ONE

Additional Editor Comments:

please ensure that your review adheres to the PRISMA guidelines and revise it accordingly.

Reviewers' comments:

Reviewer's Responses to Questions

**Comments to the Author**

1. Is the manuscript technically sound, and do the data support the conclusions?

Reviewer #1: Yes

Reviewer #2: Yes

Reviewer #3: Yes

2. Has the statistical analysis been performed appropriately and rigorously? 

Reviewer #1: N/A

Reviewer #2: N/A

Reviewer #3: Yes

3. Have the authors made all data underlying the findings in their manuscript fully available?

Reviewer #1: Yes

Reviewer #2: Yes

Reviewer #3: Yes

4. Is the manuscript presented in an intelligible fashion and written in standard English?

Reviewer #1: Yes

Reviewer #2: Yes

Reviewer #3: Yes

5. Review Comments to the Author

Reviewer #1: 1. Please explain why Lorenzo and Carrisi (2020) is included in this SR? Seems this paper is not talking about mental health / stress related issue.

2. Please provide AXIS total scores or percentage scores of each included papers to support your comment that most of the studies included in this review met the criteria for high-quality research.

Thank you.

Reviewer #2: I am pleased to review manuscript ID PONE-D-24-09879. I appreciate the authors' remarkable work. However, I have provided some critical comments below to help improve the paper.

Best of luck,

Your Reviewer

Comments and Recommendations

General Comments

• Title and Abstract:

o The title is somewhat confusing, especially when referencing "before and during COVID-19." This suggests the study is solely about COVID-19 impacts post-2020, yet the search includes studies from 1980 and papers from 2005 onward. Consider adding a timeline in the title to align with the study results as per PubMed.

o The abstract is clear and well-organized.

Methods

o Please add the heading of Study Design and Registration for the first paragraph of method.

o Please consider cite the PRISMA checklist guidelines in the method section.

• Study Characteristics:

o Include details on the highest and lowest sample sizes in the included studies, total number of participants, mean age, and age range from the youngest to oldest participant.

• Tables:

o Table 2: Characteristics of the Included Studies:

Add the first author’s last name and "et al." for studies with three or more authors.

Specify the specialties of HCWs (e.g., nurses, doctors, dentists) as you have only mentioned HCWs in general for some studies.

o Table 5: Measures Used:

Standardize the use of acronyms (e.g., GAD) and provide full terms and detailed explanations in the Result section as suggested in my following comment.

o Avoid redundancy maybe by removing the "main findings" column from Table 2 or justify its presence to prevent confusion and duplication with Table 5.

Results and Discussion

• Paragraph on Measures Used:

o Include a detailed paragraph in the results section summarizing the various measures and diagnostic tools used across the included studies.

• Strengths and Limitations of Included Papers:

o Clarify the point about the literature gap regarding research on partners and spouses in same-sex relationships, parents, and siblings of HCWs to ensure readers understand the context clearly.

Reviewer #3: For the title I recommend

"Impact of Occupational Stress on Healthcare Workers' Families Before and During COVID-19: A systematic review".

In the first paragraph of the introduction you mentioned statistics for healthcare workers (HCWs) in the UK in 2023, while you motioned statistics globally for 2020?

Use the passive voice.

6. PLOS authors have the option to publish the peer review history of their article (what does this mean?). If published, this will include your full peer review and any attached files.

Reviewer #1: No

Reviewer #2: **Yes: **Darya Rostam Ahmed

Reviewer #3: No

---

## [Author Response · Author response to Decision Letter 0]

1 Jul 2024

Thank you to you and your reviewers for your feedback on our manuscript. We are very pleased to submit a revised manuscript incorporating the reviewers’ suggestions. We have provided a point-by-point response to how we have addressed each comment in the "Response to Reviewers" letter. 

Kind regards

---

## [Editor Report · Decision Letter 1]

4 Jul 2024

PONE-D-24-09879R1Impact of Occupational Stress on Healthcare Workers' Family Members Before and During COVID-19: A systematic reviewPLOS ONE

Dear Dr. Tekin,

Thank you for submitting your manuscript to PLOS ONE. After careful consideration, we feel that it has merit but does not fully meet PLOS ONE’s publication criteria as it currently stands. Therefore, we invite you to submit a revised version of the manuscript that addresses the points raised during the review process.

We look forward to receiving your revised manuscript.

Kind regards,

Sirwan Khalid Ahmed

Academic Editor

PLOS ONE

Journal Requirements:

Additional Editor Comments:

Dear Authors,

Please make the following revisions to the manuscript based on the suggestions provided:

1. In the Study Characteristics section, please include citations in square brackets [ ] for each individual study conducted both before and after the COVID-19 pandemic. Additionally, include citations for surveys and interviews, years of publication, and for each country separately. This will help to clearly identify the specific studies that were included in your review.

2. Please provide citations in square brackets [ ] along with the names of the authors in Tables 2 and 5.

3. Differentiate between qualitative and quantitative studies in Table 2. It may be beneficial to create an additional table specifically for this purpose. Furthermore, please mention this distinction in the main text.

4. The text includes references from citation number 51 to 66. Authors, please clarify the purpose of including these various citations. It is important to appropriately reference the articles included in this review (for scales). Additionally, you may use citations 51 or 66 for further clarification.

5. Add the AMSTAR checklist, for ensuring  level of  your paper as a supplementary file. Also mention in the method section.

---

## [Author Response · Author response to Decision Letter 1]

9 Jul 2024

Good evening, 

Based on the reviewer's request, we have included the DOI numbers for the included studies in the references.

I hope this revision can address your requests. 

Kind regards

Sahra~

---

## [Editor Report · Decision Letter 2]

17 Jul 2024

Impact of Occupational Stress on Healthcare Workers' Family Members Before and During COVID-19: A systematic review

PONE-D-24-09879R2

Dear Dr. Sahra Tekin,

We’re pleased to inform you that your manuscript has been judged scientifically suitable for publication and will be formally accepted for publication once it meets all outstanding technical requirements.

Kind regards,

Sirwan Khalid Ahmed

Academic Editor

PLOS ONE
---

## [Editor Report · Acceptance letter]

23 Jul 2024

PONE-D-24-09879R2 

PLOS ONE

Dear Dr. Tekin, 

I'm pleased to inform you that your manuscript has been deemed suitable for publication in PLOS ONE. Congratulations! Your manuscript is now being handed over to our production team.

Kind regards, 

on behalf of

Dr. Sirwan Khalid Ahmed 

Academic Editor

PLOS ONE